# On Hair Care Physicochemistry: From Structure and Degradation to Novel Biobased Conditioning Agents

**DOI:** 10.3390/polym15030608

**Published:** 2023-01-24

**Authors:** Catarina Fernandes, Bruno Medronho, Luís Alves, Maria Graça Rasteiro

**Affiliations:** 1University of Coimbra, CIEPQPF, Department of Chemical Engineering, R. Sílvio Lima, 3030-790 Coimbra, Portugal; 2MED–Mediterranean Institute for Agriculture, Environment and Development, Universidade do Algarve, Faculdade de Ciências e Tecnologia, Campus de Gambelas, Ed. 8, 8005-139 Faro, Portugal; 3FSCN, Surface and Colloid Engineering, Mid Sweden University, SE-851 70 Sundsvall, Sweden

**Keywords:** biobased conditioning agents, conditioners, hair care, hair structure

## Abstract

Hair is constantly exposed to various adverse external stimuli, such as mechanical or thermal factors, that may cause damage or cause it to lose its shine and smooth appearance. These undesirable effects can be minimized by using hair conditioners, which repair the hair and restore the smooth effect desired by the consumer. Some of the currently used conditioning agents present low biodegradability and high toxicity to aquatic organisms. Consumers are also becoming more aware of environmental issues and shifting their preferences toward natural-based products. Therefore, developing novel, sustainable, natural-based derivatives that can act as conditioning agents in hair care products and thus compete with the traditional systems obtained from non-renewable sources is highly appealing. This paper presents the key physicochemical aspects of the hair conditioning process, including hair structure and degradation, and reviews some of the new alternative conditioning agents obtained from natural resources.

## 1. Introduction

Hair is characteristic of mammals. It has a protective function as it acts as a barrier for the external elements and plays a role in thermoregulation. In humans, hair also has a social significance and can cause a great impact on personal body image and confidence. Healthy hair offers emotional and psychological health and contributes to improved self-esteem and attractiveness [1,2]. The appearance of shiny, healthy, and smooth hair is dependent on environmental exposure, applied hair treatments, and simple daily hair care habits. A great variety of hair care products are available on the market to allow the consumer to achieve the desired properties and aesthetic results for their hair, such as shampoos, conditioners, dyes, mousses, lotions, etc. Other chemical and thermal treatments are also applied to change the properties of the hair, for example, straightening, curling, and bleaching processes. These treatments are preferentially performed to improve hair’s appearance and, consequently, self-image.

A crucial step in hair care is the cleaning process. Hair cleaning is provided by shampoos. These products are responsible for the removal of dust, residues from styling cosmetics, and excessive sebum accumulated in the scalp and hair fibers by solubilizing them through the surfactants present in the formulation. The soluble particles can then be easily removed by rinsing with water. Although shampoos have the purpose of contributing to and promoting well-cared-for hair, they can also leave the hair with a dry and rough appearance. The high cleaning ability of anionic surfactants present in the shampoos combined with the abrasion promoted by scrubbing the head and hair, often results in the removal and/or degradation of lipid and keratin [3]. Due to this, some people feel the need to apply hair conditioner after shampooing to improve the hair’s softness and gloss and reduce frizz.

To better understand the conditioning process, it is necessary to be familiar with the structure and composition of hair. This is especially relevant for the development of conditioning formulations because hair fibers will act as the substrate for the deposition of conditioning agents.

## 2. Hair Structure and Chemical Composition

Hair is a fibrous composite biomaterial generated by the hair follicle and composed of proteins, lipids, water, and other minor compounds, such as pigments. Proteins are the main component of hair and can account for ca. 65 to 95% of the total hair weight [4]. The main constituent of hair is keratin, a resistant fibrous protein mainly composed of tyrosine, glycine, and cysteine amino acids. The high content of sulfur-containing cysteine results in the establishment of disulfide bonds between neighboring keratin chains, forming cystine units (Figure 1) [5]. Along with disulfide bonds, peptide bonds are also abundant in the structure of the hair. Consequently, numerous −CO and −NH groups are accessible to form hydrogen bonds between adjacent proteinaceous filaments [6]. The resulting crosslinked structure of keratin is responsible for the shape, structure, and strength of the hair fibers [7,8].

In a cross-section view, the hair shaft fibers present three main concentric layers: the medulla, cortex, and cuticle, from the innermost to the outermost layer, respectively. A generic representation of the hair fiber structure is presented in Figure 2.

The medulla is a loosely packed and disordered region located at the center of the fiber and is composed of a nucleus of central cells and air spaces [2,5]. It varies among different hair types, being more noticeable in thick hair, as, for example, in a man’s beard [1,2]. Medulla represents a small part of the hair’s total weight, and its contribution to the mechanical properties of hair fibers is negligible [9]. Surrounding the medulla is the hair major component, the cortex. The cortex is composed of elongated cortical cells that contain alpha-helical keratin filaments aligned with the longitudinal axis of the hair shaft and an amorphous matrix composed of sulfur-rich proteins, mainly keratin [5,10]. In contrast to the medulla, the cortex represents ca. 90% of the total hair weight and is primarily responsible for the mechanical properties of the hair [5,11]. The filamentous structure of the cortex, with adjacent keratin filaments linked by covalent disulfide bonds, imparts strength to the hair, whereas its elasticity is provided by the helical shape of the keratin chains [2,10]. This layer also has a role in storing moisture and holding the pigments that confer color on hair [2,6]. The outer part of the hair shaft is the cuticle. This layer is composed of ca. 6–10 flat overlapping cells pointing towards the tip end of the hair fiber, resembling scales or roof shingles [12]. Each cuticle is composed of several sublayers, i.e., the epicuticle, A-layer, exocuticle, endocuticle, and the cell membrane complex [6]. A schematic illustration of the complex structure of the cuticle cell is presented in Figure 3a.

The outer component of the cuticle, the epicuticle, is covered by a lipidic layer that grants a hydrophobic character to the surface of the hair. This layer, known as the F-layer, is composed of 18-methyleicosanoic acid (18-MEA), which is covalently bonded to the cysteine groups of keratin proteins by thioester linkages (Figure 3b) [14]. The F-layer is the first hydrophobic “defensive barrier” of the hair, and it decreases the friction between hair fibers [10,15].

The cuticle acts as a barrier that protects the cortex of the fiber and is responsible for the texture and smooth appearance of hair [10]. In healthy hair, the cuticle can last up to 6 years [16]. When the hair is exposed to harsh treatments, such as chemical, physical, or thermal treatments, the hair weathering is accelerated, and the hair shaft will be damaged. The deterioration of the hair shaft starts with the removal of the F-layer, but further damage can result in the destruction of the entire cuticle and, ultimately, the degradation of the proteins in the cortex [16].

Ethnicity is a relevant factor to consider when studying hair and developing new hair care products as its structure and properties vary within individuals of different ethnicities. In this respect, hair can be categorized into three main groups: Caucasian, African, and Asian. Differences in chemical composition, structure, and properties have been studied for a long time [17,18,19,20,21,22]. The chemical composition of hair is reported to be independent of ethnic origin; all hair has the same chemical properties, i.e., protein composition [17,18]. Regarding the amount of fibrous proteins (FPs) and matrix substances (MSs), these were found to not vary among subjects within the same ethnic group, but differences were observed between groups [19]. The yields of FPs from Asian hair were significantly higher than those of African and Caucasian hairs. Regarding MSs, higher yields were observed for African hair than for the other two groups. Nevertheless, these parameters do not show significant differences within all three groups. More interesting is the comparison of the ratios of FPs and MSs between ethnic groups. When accessing this new parameter, it was found that all three groups have significantly different ratios. Asian hair has the highest ratio of FPs to MSs yields, while African hair is on the other side of the spectrum, with the lowest value. The variation of the content of fibrous proteins of different hair types was suggested to be related to its macroscopic properties [19]. The main distinguishing features among hair types are the shape and diameter of the hair shaft. Asian hair is generally thicker, with a diameter of ca. 100 μm and a rounded cross-section shape, while Caucasian hair is usually thinner, with a diameter of ca. 50 μm and a slightly elliptical shape [22,23]. The African hair has an intermediate thickness of ca. 80 μm, although with a higher heterogeneity in diameter than the previous ones and an elliptical cross-section [20,23]. The visible cuticle cells length also varies from 5 to 10 μm according to the hair type [23]. These features can be observed in the SEM images of hair samples from the three ethnic groups presented in Figure 4. Another distinctive feature is the hair’s shape: African hair presents a characteristically highly curled structure, while Asian hair is usually very straight. On the other hand, Caucasian hair can be straight, wavy, or slightly curly [21]. The cross-sectional shape and mechanical properties of the hair shaft can be attributed to the previously mentioned ratio between fibrous protein and matrix contents. The higher degree of fibrous proteins relatively to matrix observed in Asian hair may be responsible for the circular cross-section and higher strength observed in those fibers, while the flattened and more susceptible to breakage African hair may result from the lowest ratio of fibrous proteins [19,21].

Understanding the differences in structure and properties in the hair of subjects of different ethnicities, such as the mechanical behavior, shape, and damage mechanism, is of utmost importance for the development of efficient hair care formulations tailored to the desired applications and fulfilling the consumers’ expectations. In addition, the reproducibility of the results depends on the similarity of the properties between samples. Therefore, it is important to take into consideration the possible variations in hair’s morphology when conducting different experiments [8].

### 2.1. Hair Weathering

Hair weathering is the chemical or physical deterioration of the hair shaft that results in the removal of some hair components and makes it more susceptible to breakage [3]. The damage is caused, for example, by cosmetic procedures, bad hair care habits, or environmental factors such as prolonged exposure to UV radiation [24]. Some cosmetic treatments are extremely harsh and result in increased structural and chemical damage to the hair fibers. Chemical procedures, such as bleaching, perming, relaxing, and straightening, are known to cause significant damage to hair fibers [16]. Nonetheless, inadequate daily care routines and practices are also a cause of hair deterioration. Physical damage can result from poor brushing or combing or from blow drying at extremely high temperatures [16]. Hair weathering is a progressive process of deterioration where the hair shaft loses the external protection provided by the cuticles and affects the ability of the cortex to retain moisture [24]. In a first stage, this results in loss of gloss and softness, but with further damage, hair may lose its strength and elasticity, and eventually its complete structural integrity, resulting in hair breakage [25]. Hair is constantly exposed to various sources of weathering during its existence. This means that older hair will present a higher degree of damage than the recently formed hair shaft. Consequently, the hair near the scalp presents a more preserved structure than the tips of long hair. In Figure 5, examples of hair samples showing remarkable injuries on the hair shaft along the fiber’s length are presented. Near the scalp, the hair has complete cuticles, and the hair shaft is fully covered by the overlapping scales. As the distance from the scalp increases, the hair becomes more damaged. Some cuticles were partially removed, exposing the cortex of the fiber, and the surface shows a rough appearance. Near the tip (Figure 5, right), no cuticles are visible due to the longer exposure to external insults during its existence, which resulted in the complete removal of the cuticle layers. The degree of damage will differ from person to person, and it is strongly dependent on the daily care habits of each person and the degree of exposure to weathering events.

Deterioration of the hair fibers occurs through four categories of damage causes: mechanical, chemical, thermal, or environmental processes, or a combination of those. The degree of damage caused by each type of external insult will depend on the habits of each individual, but, in general, chemical processes are the ones prone to inflict more deterioration, whereas exposure to environmental factors has a lower impact [16].

#### 2.1.1. Mechanical Insults

Mechanical damage is the most frequent damage source because it occurs every day when, for example, combing the hair with plastic or wood combs or brushes. The repeatedly brushed movements in the same area cause the cuticle edges to peel off and, ultimately, to be partially or totally removed, especially on the outside edges of the fiber.

Cutting the hair is another form of mechanical insult, especially when not performed with appropriate tools. The use of razors instead of scissors or even a blunt scissor will result in a cut that is not clean, and the cuticle will be more prone to peeling off [16].

Although shampoos have the purpose of promoting cleaning and maintaining the hair in good condition, the shampooing action itself can be a source of damage to the hair causing deformation and abrasion/erosion when bending the fibers and scrubbing the hairs against each other. Moreover, the high cleaning ability of the anionic surfactants in shampoos can also promote the dissolution of structural lipids and proteins [3].

#### 2.1.2. Environmental Insults

The pigment molecules, i.e., melanin, present in the hair shafts grant them photoprotection against UV radiation by absorbing and filtering it [26]. This filtering partially avoids the degradation of the proteins by radiation, but the pigments are degraded or bleached during the process, changing the appearance of the hair. For the reason that the cuticles are exposed to higher intensities of radiation and melanin is only located in the cortex, they are more vulnerable to photodamage [1]. Sun exposure induces modifications in amino acids, especially in the cuticles, resulting in rupture and detachment of the external layers, causing the splitting of the ends. It also causes the hair to become dry and brittle, lose its luster and color, and have a rough surface [26]. The photochemical degradation of proteins results in a decrease in their structural integrity. This occurs by the rupture of disulfide bonds between structural units and the establishment of new bonds by the reaction of carbonyl groups with amino groups within and between structural units [27]. Independent of the hair type, the UVB radiation is the major cause of hair protein loss, whereas the UVA radiation is responsible for color changes [26]. Even though the type of damage inflicted on the hair is only dependent on the radiation wavelength, the degree of damage varies according to the hair type. Different types of hair have different photostability due to their differences in melanin type and concentration. Black hair seems to be more photostable than blond hair and presents higher protection against UV light in the melanin-rich cortex, showing only a slight modification of fiber proteins under irradiation [27]. When comparing the photodamage in the cuticles, the protein loss is similar between black and light-brown hairs, which is explained by the absence of melanin at the surface of the fibers [28]. Changes in hair color also depend on the type of hair, being more pronounced in light-colored hair than in black hair [26].

#### 2.1.3. Chemical Insults

Chemical procedures for hair styling, such as chemical straightening, bleaching, or dyeing, are responsible for severe injuries to the hair, especially when not performed by well-trained professionals. For example, chemical straightening is a chemical procedure adopted as a long-lasting alternative to thermal straightening. It is performed with chemical relaxers that produce a straight appearance by affecting the cysteine disulfide bonds of the hair [29]. Along with the rearrangement of disulfide bonds, structural damage to the hair shaft also occurs due to modifications in keratin’s linkages, resulting in weaker hair with reduced tensile strength [11,29].

Bleaching is the process that allows for the lightening of the hair by oxidizing melanin pigments present in the cortex. This process disrupts disulfide bonds and makes the cuticles more porous, leading to poorer tensile strength and a brittle appearance [1]. The photoprotective function of melanin is lost when it is oxidized, so bleached hair is more susceptible to being further damaged by UV exposure [1].

Permanent dyes are used to change the color of the hair fibers. Similarly to bleaching, their effect is also based on oxidation reactions that occur within the hair shaft. To enable their penetration into the hair shaft, alkaline solutions (e.g., ammonia) are applied to disrupt the protective barrier of the cuticle, making hair especially susceptible to future damage [1]. Application of alkaline treatments results in the removal of the 18-MEA layer, increasing the hair’s hydrophilicity and friction [15]. The loss of this hydrophobic layer reduces the smooth appearance of hair and makes it more fragile.

#### 2.1.4. Thermal Insults

The use of blow-dryers to dry the hair is a common practice. However, its regular use causes damage to the hair shaft. While air-dried hair shows a well-protected surface, blow-dried hair shows a higher degree of damage [30]. The hair surface becomes more damaged as the temperature increases [30]. Blow-drying the hair causes the scale edges to become concave-shaped due to cuticle layer splitting [31]. Structural changes caused by blow-drying are not exclusively limited to the surface of the fibers but also occur between inner cuticle layers [31]. Hair color is also affected by heat, becoming lighter after repeated shampooing and drying. These changes in color are visible after just 10 blow-drying processes at 95 °C [30].

Styling the hair with hot straightening or curling irons also inflicts damage to the hair. These processes occur at very high temperatures, and the styling effects are caused by a temporary rearrangement of the hydrogen and disulfide bonds within the hair shaft [11,29]. Thermal straightening promotes overheating of the hair shaft, causing weakening and rupture [29].

## 3. Hair Care

People care about their health and appearance. A beautiful, healthy and well-treated hair and skin causes impact on the others and self-impression of one’s image. Since way back, different products have been used to change hair, skin, and physical features. These goods are often grouped as cosmetic products, and, according to the proper EU legislation (Regulation No. 1223/2009), these are described as “any substance or mixture intended to be placed in contact with the external parts of the human body (epidermis, hair system, nails, lips, and external genital organs) or with the teeth and the mucous membranes of the oral cavity with a view exclusively or mainly to cleaning them, perfuming them, changing their appearance, protecting them, keeping them in good condition or correcting body odors” [32].

Hair has a great impact on a person’s image and expression of personality. Some practices, namely those procedures performed in hair salons, such as cutting, dying, or styling the hair, are implemented with the fundamental purpose of embellishing. On a daily basis, some care practices also help to improve hair’s feel and look, such as grooming habits. Brushing, cleaning, and the application of cosmetic products play a major role in accomplishing these results. These practices are of extreme importance to keeping the hair and scalp clean, maintaining the hair in good condition, and repairing or, at the least, mitigating the damage caused by external aggressions. For the reason that hair is composed of dead cells, it cannot be biologically renovated; the only way to restore its properties is to physically repair it by using a hair conditioner and/or other hair care products that can cover or fill the damaged surface of the hair shaft [23,33].

## 4. Hair Conditioners

Environmental, thermal, and chemical weathering cause damage to the hair surface, resulting in an unpleasant feeling and appearance. The primary reason for luster and texture loss is the partial or total removal of the 18-MEA layer from the cuticle surface. The lack of this hydrophobic protection makes the hair more hydrophilic and susceptible to swelling, thus increasing its fragility and the friction between the fibers [10,15]. As mentioned, hair fibers are composed of dead cells that cannot replicate, and thus hair cannot be biologically repaired [5]. To avoid the exposure and damage of the hair’s cortex, the surface of the fiber can be covered and partially repaired by applying hair conditioner that fills the damaged area of the cuticle [5]. In Figure 6, SEM images of virgin hair and hair treated with conditioner are presented, showing the deposition of hair conditioner that fills the edges of the cuticles. When using intensive conditioners, the F-layer can be temporarily replaced, and it is possible to restore some of the lost physical properties of hair [34].

Hair conditioner is one of the numerous hair care products available. The main function of conditioners is to improve the physical and aesthetic properties of the hair. They are primarily designed for dry and damaged hair and contain molecules that build up on the damaged scales of the cuticle, filling the fractures and fissures [35]. This results in a smooth surface with reduced friction between hair filaments. Briefly, hair conditioners impart softness, gloss, and luster to the hair, reduce frizz, and improve its manageability and ease of combing [5,34,35].

### 4.1. Classification of Hair Conditioners

There are various cosmetic approaches to imparting hair conditioning. Hair conditioners can be categorized in different ways, according to the type of product (i.e., the way it is applied to the hair), the action mode, or its composition. The most suitable conditioner formulation is dependent on the desired effect and damage degree. The differences between the various products will be presented in more detail in this section.

Regarding the type of formulation, hair-conditioning agents can be incorporated in shampoos or used in the form of post-shampooing lotions, mousses, or gels. In shampoo formulations, conditioning agents are included as additives to prevent or reduce the negative effects promoted by anionic surfactants, which increase the electrical negative charges on the hair surface and create frizz and friction. They are designed for dry and damaged hair to impart softness and gloss and diminish hair entanglement [36]. Although some degree of conditioning can be achieved with the incorporation of conditioning additives in shampoo formulations, their effect is not as pronounced as in a hair conditioner formulation, possibly due to incompatibilities between the components, in particular, anionic surfactant cleaners and cationic conditioners, which may react with each other and, as a result, create complexes that are insoluble in water [36]. In “2 in 1” shampoos, the formulation is intended to have a dual function: cleaning and conditioning. Thus, it is expected that these products are not as powerful as the ones designed specifically for that purpose. On the other hand, post-shampoo conditioners are formulated with the specific purpose of imparting conditioning to the hair and, therefore, are much more effective. Some of them are supposed to be applied for a certain period and then rinsed off (i.e., instant and deep conditioners), while others are designed to be left in the hair (i.e., leave-in and blow-drying conditioners). Instant conditioners are the most common and are generally applied after every hair wash. They are applied to wet hair, left on for about 5 min, and then rinsed off with water. Due to the short contact time, instant conditioners are not the ones that impart the best conditioning, but they are effective at reducing the dryness caused by shampooing and other daily damages, thus helping to improve hair manageability. The detangling of the hair occurs by smoothing the cuticles, causing a reduction in friction when applied to slightly damaged hair [37]. If the hair is severely injured and dry, it is recommended the use of deep conditioners because they are more concentrated than the instant conditioners and they are meant to be left in the hair for longer periods, usually 20 to 30 min [38]. Additional application of heat results in the cuticles’ lifting and increases the penetration of the conditioning agents [33,39]. For a longer conditioning effect, consumers often use leave-in formulations in addition to or instead of rinse-off products. These products are formulated to be applied to wet or dry hair before blow-drying or styling. They usually contain silicones or other lightweight polymers, such as polyvinylpyrrolidone, that form a thin film that coats the hair surface and reduces static electricity [33]. Leave-in and blow-drying conditioners are useful to prevent the damage caused by daily care routines and to grant thermo- and photoprotection [40].

In Table 1, the main characteristics and the active ingredients commonly used in the formulation of the various types of conditioners are summarized.

As shown in Table 1, there is a wide variety of products with conditioning properties, and, even for each type of product, a wide variety of compounds can be introduced to impart condition to the hair. Depending on the severity of the damage and the aesthetical and physical properties required by the consumer, different classes of compounds with diverse behaviors and modes of action are used to design tailored hair conditioners for each type of hair and damage degree. There is a plentiful list of compounds with conditioning properties. Some of them include natural polymers (e.g., cellulose, starch, and their derivatives), synthetic polymers (e.g., polyvinylpyrrolidone, polyacrylates, silicones, etc.), gums, and hydrolyzed proteins [41]. The interactions of the conditioning compounds with hair are affected by the polymer’s properties, for example, its molecular weight, hydrophobicity, and charge density. So, it is expected that the different classes of conditioning agents will act differently and grant different properties to the hair. Hair conditioning is achieved by at least one of the three key processes presented below:Neutralization of the negative charges of the hair fibers through the adsorption of cationic compounds onto the surface.Lubrication of the cuticles by restoring the hydrophobic character of the hair shaft.Restoring the lost proteins and enabling moisture retention through treatment with small proteins that penetrate the hair shaft.

Each one of these pathways to achieving smooth, shiny, and manageable hair is accomplished by using compounds with different chemical natures. Hair conditioners can be divided into three classes according to their composition and mechanism of action: cationic, film-forming, and protein-based conditioners. An overview of the main characteristics of each type of conditioning agent and examples of compounds used in the formulations are given in Table 2.

Cationic conditioners are characterized by the presence of cationic surfactants, also known as quaternary ammonium compounds. Due to their positive charges, this type of conditioning agent neutralizes the negative charges of the hair, promoting a decrease in static electricity [42]. The thin film of conditioner that adsorbs at the hair surface flattens the cuticles, making them smooth and improving the softness and shine of the hair. Cationic surfactants are very effective for the conditioning of chemically treated hair because of their high density of negative charges. Several quaternary ammonium salts are used as conditioning agents in hair care formulations, such as cetrimonium chloride, behentrimonium chloride, stearalkonium chloride, etc. These compounds have a hydrophilic, positively charged group that interacts with the hair shaft through electrostatic interactions and one or more hydrophobic alkyl chains that point towards the outer surface, which enables them to restore some of the lost hydrophobicity (Figure 7). It also helps to improve the compatibility between hair and other compounds in the formulation that synergistically contribute to the conditioning effects (such as oils) by changing the hydrophilic character of the hair’s surface [43].

Film-forming conditioners are prepared from high-molecular-weight polymers that form a thin film, coating the surface of the hair fiber while filling the cuticle defects. This coating smooths the cuticle’s surface and reduces the friction between hair shafts. Consequently, the coated hair will present lower static electricity and reduced frizz. Silicones are one of the most common film-forming agents used in conditioners. There are different types of silicones that show different deposition, adherence, and rinse-out capacities, resulting in conditioners with distinct performances [45]. Dimethicone is the most commonly used within the silicones family [10]. It is a hydrophobic polymer, so it will contribute to restoring the hydrophobicity of the hair and act as a lubricant. However, the adsorption of hydrophobic polymers will be greater in virgin hair with limited damage. To promote the deposition of dimethicone on anionic fibers, additional cationic bridging agents are added to the formulation to increase the affinity between charged hair and silicone [45]. An alternative approach is to add cationically-modified polymers that combine the film-forming action of high molecular weight polymers with the enhanced interactions of cationic surfactants and consequent neutralization of hair’s surface charge. Cationic polymers, such as Polyquaternium-10 (quaternized cellulose) and Polyquaternium-7 (co-polymer of diallyl dimethyl ammonium chloride and acrylamide), are commonly used, but more in shampoo formulations than in conditioners [3].

Conditioners containing proteins in their composition are efficient at improving the health of the hair by temporarily repairing its damage, particularly at the split ends. Hair weathering induces the removal of external cuticle scales and leaves the hair fragile and susceptible to breaking or splitting. Protein-based conditioners are composed of low molecular weight (i.e., 1 to 10 kDa) hydrolyzed protein fragments, such as amino acids or small peptides, that can penetrate the hair shaft, bind to the keratin, and restore the lost proteins [10]. This greatly improves hair strength and helps to avoid further damage. Hydrolyzed proteins are obtained from various sources, such as animal collagen, keratin, placenta, etc. [46]. The efficiency of the conditioner is not as dependent on the protein source as it is on its particle size and its consequent ability to penetrate and remain inside the hair shaft [46]. The contact time is also an important factor: to achieve a greater effect, the conditioner should be left in contact with the hair for a longer period, thus allowing the proteins to diffuse into the fibers. The effect of protein conditioning is only temporary: the excess proteins attached to the hair are removed when shampooing, and thus it is necessary to reapply the conditioner to maintain the conditioning effect [33].

Most hair conditioner products are formulated with a mixture of different conditioning agents to enhance their performance. High concentrations of fatty alcohols, such as cetyl and/or stearyl alcohols, or other lipid components are found in hair conditioners in addition to cationic surfactants [3]. This is beneficial because hydrophobic lubricants, such as oils and silicones, are not as compatible with damaged hair as cationic compounds. When incorporating cationic surfactants in addition to these film-forming agents, they will act as binding bridges between highly polar, negatively charged hair and non-polar conditioners. As previously shown in Figure 7, cationic surfactants interact with the hair by its cationic polar group, leaving a thin film of hydrophobic alkyl chains pointing to the outer surface that is a compatible substrate for further deposition of hydrophobic compounds.

### 4.2. Physicochemical Principles of the Hair-Conditioning Process

As discussed above, various factors can influence the structure and aesthetic properties of hair. Some of them cause severe damage to the hair fibers, such as chemical treatments, but even environmental weathering alters the hair’s surface and integrity, leaving the hair with an unhealthy feeling and appearance. These alterations may be caused, for example, by the removal of the outermost lipidic 18-MEA layer that is covalently linked to the cuticle or by the oxidation of the disulfide bonds from the cystine residues to cysteic acid caused by prolonged sunlight exposure [5]. The excess of acidic groups undergo dissociation in the aqueous medium and increase the charge density in the hair shaft [5]. These modifications change the character of the hair surface from highly hydrophobic and poorly charged to a hydrophilic, polar, and negatively charged surface [3]. It was stated that the use of conditioners is effective to prevent or treat these undesired modifications. The performance of the hair conditioners is dependent on their composition but also on their ability to deposit onto the hair surface and remain attached to it. Therefore, understanding the interactions between conditioner ingredients and hair is of extreme importance, since they will affect the distribution, thickness, and, consequently, the proper function of conditioners [47].

As previously discussed, the variety of hair conditioners is large, and their different compounds act in a distinct manner due to the different interactions that govern their deposition or absorption into the hair. For example, hydrolyzed proteins of small size can diffuse into the hair shaft, whereas polymers of high molecular weight only act at the cuticle surface by adsorbing via van der Waals interactions or binding via chemical or electrostatic interactions [47]. For the reason hair conditioners are aqueous solutions or suspensions, the amount of product that deposits onto the fiber depends on the balance between its interactions with the keratin, its hydrophilicity or binding forces to the aqueous phase, and its diffusion rate [3]. Adsorption of different compounds to the hair keratins is considered a continuum spectrum of mechanisms between charge-driven and hydrophobically-driven processes [48]. Some compounds, such as water-soluble cationic surfactants, at a pH above the isoelectric point of the hair proteins, adsorb onto their surface essentially by a charge-driven mechanism due to the attraction between positively charged quaternary ammonium ions and negatively charged hair [3]. In contrast, the adsorption of water-insoluble dimethicone from an anionic shampoo medium onto the hair surface is essentially driven by hydrophobic interactions [3]. These examples are representative of the extreme ends of the spectrum. However, subtle structural changes in the adsorbing compounds or in the medium conditions, such as pH alteration, may induce changes in the adsorption mechanism towards an intermediate mixed process with contributions of both charge and hydrophobic interactions. The interactions with keratin are affected by various factors, such as the molecular size and charge of the conditioning agents, the pH of the medium, and the hair’s properties, such as damage degree and isoelectric point [3]. The isoelectric point and pH of the medium affect the mechanism of deposition of the different compounds because these factors are directly related to the hair’s net charge. The isoelectric point of hair is acidic, around 3.67 [49]. This means that above this pH, the hair will present a net negative charge density, and cationic conditioners are prone to electrostatically interact with it. In the case of damaged hair, higher amounts of conditioners are adsorbed due to their lower isoelectric point, which increases the concentration of negative charges at the surface. The deposition of cationic conditioner onto anionic hair cuticles is illustrated in Figure 8. In brief, the conditioner deposits flatten the cuticle scales against each other, reducing the flyaway behavior and improving the hair’s shine and color [50].

### 4.3. Alternative Conditioning Agents

Some of the conventional hair conditioning agents are reported to cause skin and eye irritation and adverse effects on the natural environment, as they are toxic to aquatic organisms with long-term effects [51]. Due to this, the development of highly efficient conditioners with low impact on human health and the environment is highly desirable. Consumers’ awareness regarding environmental issues is growing, thus driving their preferences towards sustainable and natural-based formulations. Research on the assessment of the public perception of bio-based products and the bioeconomy has been conducted in the past few years in several projects, such as BIOWAYS [52] and RoadToBio [53]. Results of surveys clearly show that the vast majority of the respondents (80−82%) express a positive overall perception of bio-based products [54,55]. The public perceives the advantages of using bio-based products, especially regarding environmental benefits in terms of land, water, and biodiversity (83.6%) and less use of fossil fuels (74.5%) [56]. In addition, a significant number of respondents (64.2%) are confident that the use of bio-based products contributes to the creation of sustainable economic growth and new jobs [54]. Regarding their preference for biobased or conventional products, 66.6% of participants would prefer biobased products over non-biobased products, if available [56]. The majority of respondents (50.2%) believe that biobased products are just as good as conventional ones, and 53.1% answered that they are willing to pay more for a biobased product with the same functionality and properties as a fossil-fuel-derived one [54]. Despite their positive perception and preference, only 12% of consumers have ever consciously chosen biobased products over conventional ones [55], and 71% of the respondents stated that they never use biobased products [54]. This could be related to their lack of awareness or difficulty finding biobased products since only 25.6% agree that they can easily find biobased products on the market [54]. From these results, it is clear that people are interested in biobased products, but more biobased solutions must be available on the market or, at the least, well-identified so that consumers can easily find these alternatives and become aware of their benefits regarding the impacts on health and the environment. It is mainly the limited availability and the lack of labelling of bio-based products that discourage people from using them [56].

A common practice that has been adopted by the cosmetic industry to improve the sustainability of cosmetic products is to produce formulations containing natural ingredients. This approach allows the cosmetic industry to meet consumer expectations regarding the production of more eco-sustainable formulations as well as corporate social responsibility (CSR). The European Commission defines CSR as “the responsibility of enterprises for their impacts on society” and says it should integrate social, environmental, ethical, human rights, and consumer concerns [57]. Some cosmetic industries are committed to sustainable development and finding raw materials of natural origin [58,59]. The sustainable development process should consider all stages of the development process, including the life cycle of the ingredients, and should follow the green chemistry principles [59,60]. Although there is a vast variety of natural ingredients, some recommendations are described for the choice of sustainable ingredients: (1) use renewable resources; (2) use green fabrication processes; and (3) have a low environmental impact [59,60].

Despite their natural origin, it is crucial to assess the information on the compounds’ properties. Regarding cosmetic products, it is particularly important to consider their biodegradation, toxicity, and dermatological features [61]. Nonetheless, the product properties and overall efficiency should not be neglected when developing new hair care products. The goal is to create more sustainable formulations without compromising the conditioning effect, i.e., wet and dry combing forces, rheological properties, and product stability. Protein-based conditioners, for example, can be easily prepared from natural ingredients. Hydrolyzed keratin proteins from human origin or sheep wool are commonly used due to their great ability to repair and restore human hair. However, human and animal-based products face ethical and health concerns, and their use is limited by regulations [62]. Due to this, proteins obtained from vegetable sources are highly appealing raw materials for the development of cost-effective and sustainable haircare formulations [63]. Plant-derived proteins, such as those derived from quinoa, jojoba, baobab, soy, or rice, can be used as conditioning agents in cosmetic formulations [5]. Peptides synthesized from naturally derived amino acids are also proposed as efficient, sustainable conditioning agents with tunable structure and properties [62]. This route allows for fine tuning the molecular weight of these keratins to make them suitable for penetration into the hair shaft and to selecting the terminal peptides to make them more substantive to the hair [62]. It has been shown that a high level of peptide deposition can be easily achieved and maintained on hair after daily use.

Vegetable-based lipids can also act as conditioning agents and restore the hydrophobic character of the hair shaft by replacing the hair’s natural lipidic layer. Recently, Rincón-Fontán, et al. [64] studied the adsorption of a lipopeptide biosurfactant extracted from corn steep liquor (CSL) on human hair. This biosurfactant is composed of 64.2% fatty acids (mainly C18–C16 fatty acids) and 21.9% of proteins, and based on its biocompatibility, it could represent a promising ecofriendly alternative to the conventional surfactants used in hair care formulations. The adsorption of an aqueous solution containing the biosurfactant was studied based on a fractional factorial design with three independent variables: temperature (20–50 °C), pH (5–7) and treatment time (2–30 min) and the adsorption capacity of hair was set as a dependent variable. The temperature was observed to be the variable with the highest influence on the adsorption capacity, followed by the pH. The treatment time was the least significant independent variable, suggesting that the adsorption of the biosurfactant onto hair occurs very quickly, within a few minutes after the treatment. When the interactions between two variables are considered, a significant effect on the adsorption capacity of biosurfactant is observed between temperature and time and also between temperature and pH. For a fixed treatment time of 2 min, higher pH and temperature increase the adsorption capacity, whereas, at low temperature, the pH has a lower effect. For a longer treatment time (16 min), the capacity of hair to adsorb the biosurfactant was observed to be higher at lower temperatures. The values for the maximum capacity predicted by the model and experimentally obtained were 3382 μg.g^−1^ and 3679 μg.g^−1^, respectively, and are in the range of those previously reported for the adsorption of a cationic petrochemical-derived surfactant [64,65]. This biosurfactant can efficiently adsorb onto the hair surface and restore the fatty acid coating naturally present in healthy hair.

Yorke and Amin [66] prepared a new conditioning shampoo using a natural conditioning agent, hyaluronic acid (HA), in combination with a biosurfactant (synthesized by microorganisms) or biobased surfactant (manufactured from biobased sources). The formulations containing HA were prepared at a variety of molecular weights and concentrations in combination with a surfactant, either an acidic sophorolipid (biosurfactant) or an alkyl polyglucoside (biobased surfactant), and varying the presence of sodium chloride. The prepared conditioning shampoos were then characterized for their wet combing force, their viscosities at various shear rates, surface tension, and foamability. The goal was to develop a formulation containing one of the studied surfactants in combination with HA with the following characteristics: non-Newtonian, shear-thinning behavior, high foamability, and minimal wet combing force and surface tension. The authors have shown that molecular weight and concentration have a large impact on wet combing force as well as their rheological features. Regarding the reduction in wet combing force of formulations containing only HA at different molecular weights, it was observed that the highest reduction in wet combing force (85%) is achieved with the lowest molecular weight HA (8 kDa), being the formulation with the best conditioning effect. The reduction dropped to 79.5% and 49.2% when increasing the molecular weights to 90 and 130 kDa, respectively. However, a further increase in molecular weight to 800 kDa resulted in a slightly larger reduction in wet combing force (54.2%). In contrast to lower molecular weight HAs that show Newtonian fluid behavior, the largest molecular weight (800 kDa) shows the desired shear-thinning profile, and, at higher concentrations, it results in the lowest friction coefficient. Overall, a formulation containing 1 wt.% HA with 800 kDa, 10 wt.% acidic sophorolipid (biosurfactant), and 1 wt.% NaCl exhibited an excellent reduction in wet combing force, desired rheological features, and a decent decrease in surface tension with suitable foam generation. This system appears as a potential natural, sustainable, and biocompatible alternative to silicone-based products.

Extremely damaged hair, especially chemically treated hair, has a lower isoelectric point and, consequently, a higher density of negative sites [49]. Due to this, consumers with damaged hair have difficulties achieving the desired conditioning effect without using cationic conditioners [60]. However, the most common cationic surfactants are mostly derived from petrochemicals, are poorly biodegradable, and have acute aquatic toxicity [67]. Natural renewable resources are being investigated for the development of cationic conditioners with minimal impact on the environment and human health.

Esterquats are a common class of cationic surfactants that can be introduced in hair care formulations as conditioning agents. Esterquats are quaternary ammonium compounds derived from alkanolamines, mainly triethanolamine, methyldiethanolamine, or dimethylamino-1,2-propanediol and fatty acids, such as oleic or palmitic acids, from vegetable oils [61]. Esterquats containing cleavable ester bonds link the fatty acids to the quaternary ammonium head group [68]. Alkyl esterquats emerged as a more environmentally friendly alternative to conventional and more stable alkyl quats. In comparison to stable quats, the overall ecological characteristics of esterquats are superior, as they show excellent values for biodegradability and aquatic toxicity [69]. This encourages their use in several applications, including as conditioning agents in hair care formulations. Although the lower stability of esterquats caused by the presence of cleavable ester bonds is beneficial from an environmental point of view, it can limit some applications or be challenging because their stability in aqueous solutions is limited to a narrow pH range and for a certain period. Some examples of commercially available esterquats include disunfloweroylethyl dimonium chloride [70] and dioleoylethyl hydroxyethylmonium methosulfate [71], whose chemical structures are illustrated in Figure 9. These compounds are produced from plant-based raw materials, such as sunflower seed oil, via the C18 fatty chain moieties and high degree of unsaturation of this oil, which is believed to improve the conditioner’s coating and lubricating properties. Both conditioning agents have a renewable carbon index of 88% and can be used as conditioning agents and/or feel enhancers in a wide range of products, such as rinse-off or leave-in hair conditioners, hair styling products, and conditioning shampoos, among others.

Ajayi, et al. [72] conducted a study to evaluate the performance of a novel cationic amino lipid surfactant, brassicyl valinate esylate (BVE), in contrast to conventional alkyl quaternary ammonium surfactants, behentrimonium chloride (BTAC), and cetrimonium chloride (CTAC). The chemical structures of the three cationic surfactants are presented in Figure 10. BVE, also known as AminoSensyl™, is a natural and eco-friendly cationic amino acid-based surfactant derived from the essential amino acid valine and *Brassica napus* seed oil [73]. The results from the tensile combing tests revealed an overall reduction in combing force of 97.81% for the tress treated with BVE. Both BVE and BTAC performed comparably and showed better combing results than CTAC, which can be attributed to the difference in chain length of the hydrophobic tail group among the three surfactants [72]. Cationic surfactants with a longer alkyl chain usually improve the conditioning effect [44]. Thus, BVE was found to be an effective conditioning surfactant and a suitable eco-friendly alternative for BTAC and CTAC in the cosmetic preparation of hair conditioners.

Wheat proteins are mainly composed of glutenin and gliadin, which contain a high content of inter- and intra-polypeptide disulfide bonds [74]. For that reason, wheat proteins are promising conditioning agents to recover damaged hair [75]. Wang, et al. [75] studied the effect of wheat protein hydrolysates on the recovery of damaged hair when used in shampoo formulations. To improve the adherence to the hair’s keratin, cationization of wheat protein hydrolysates with epoxypropyl dodecyl dimethyl ammonium chloride was performed. Three series of conditioning shampoos containing cationic hydroxyethyl cellulose (CHEC), wheat protein hydrolysate (WPH), or quaternized wheat protein hydrolysate (QWPH) were prepared and characterized for their conditioning effect. For all the conditioning shampoos, the combing work was smaller than that for the shampoo with no conditioning agent, and it further decreased with the increase in concentration. The differences in the wet combing forces of CHEC and WPH at the same concentration were not as pronounced as those observed when using QWPH. To evaluate the attachment efficiency of the conditioners, the authors determined the weight gain percentage of hairs after treatment with the different shampoos. It was observed that for the CHEC and WPH series, the weight gain rate increased with the increase in concentration, with CHEC showing higher values than WPH for all concentrations. For QWPH, the weight gain percentage was significantly higher, and the dependence on the concentration was not observed for concentrations above 1%. The overall order of weight gain was QWPH > CHEC > WPH. As expected, the cationic nature of QWPH and CHEC facilitates their attachment to the hair. The amino acid residue of QWPH, in combination with its cationic nature, improved its binding ability, making it a suitable candidate for recovering damaged hair and smoothing the hair surface.

These examples illustrate the great variety of natural compounds that can be used to prepare eco-friendly and sustainable conditioning formulations. Some other natural sources with conditioning properties are listed in Table 3 and include, for example, animal or vegetable proteins or cellulose derivatives. Besides their conditioning effect, other advantageous properties are listed, such as their ability to act as emulsion stabilizers or thickening agents.

### 4.4. Lignocellulosic Biomass as a Platform for Hair Care Products

Some examples of natural compounds that are currently used as conditioning agents in hair care products were presented in the previous section. Cellulose was mentioned as a source of both non-ionic or cationic derivatives with film-forming and antistatic properties. Cellulose’s primary sources are lignocellulosic materials, such as wood and agroforest residues. Lignocellulosic biomass, as the most abundant natural and renewable resource on Earth, has a great potential as a sustainable supply for chemicals and biofuels production. Since lignocellulose is not part of the human food chain, its use to produce bioproducts does not threaten the world’s food supply [77]. The use of cellulose is highly appealing because it not only has a natural origin but can also be obtained from wastes and by-products of several agroforest industries. The valorization of these residues is aligned with the circular economy and zero-waste approaches. The natural ingredients obtained from agroforest wastes are reported to be, in general, biodegradable or compostable under specific environmental conditions, not harmful, biocompatible and can be used in a wide range of topical applications [78]. In addition to cellulose, lignocellulosic materials are composed of other polymers, such as lignin, that can be a valuable alternative raw material for new biobased products.

In contrast to cellulose, which is widely used in several applications, lignin is still highly underutilized, being most of it burned as low-grade fuel [79,80]. Currently, only ca. 2% of the available lignin is commercialized mainly as adhesives or dispersants [81]. However, lignin has a great potential as a sustainable alternative feedstock for the development of novel biomaterials and its valorization would expand the utilization of the biomass. As a natural polyphenol, lignin is a very attractive source of bio-phenol, being a potential sustainable substitute for petroleum-based phenol [82]. It can find applications in different areas, such as biofoams, composites and blends, food packaging, resins, flocculants and pharmaceutical and biomedical applications, showing its notable interdisciplinarity and multifaceted capacity [79]. In addition, lignin shows several valuable properties, such as biodegradability, antimicrobial and antibacterial activities, and the ability to absorb UV radiation. Due to these attractive properties and its unique structure, lignin has been investigated as a potential active ingredient in cosmetic formulations. The application of lignin as conditioning agent for hair products has not been previously described. However, several studies report lignin’s potential as a cosmetic active ingredient for both hair and skin products [11,83,84,85,86,87,88,89,90,91,92,93,94,95,96,97].

In the hair care sector, a straightening booster containing sodium lignosulfonate in its composition is commercially available [98]. This formulation was developed for the temporary realignment of hair fibers, thus reducing volume and frizz [98]. However, the specific role of lignin in the formulation is not described.

Lignin was recently proposed as a carrier for active ingredients that would contribute for the development of innovative and smart cosmetic hairline by Morganti, et al. [92]. The authors have previously developed cosmeceutical tissues composed of chitin nanofibrils (CN) and nanolignin (NL) microcapsules with the ability to bind to different active ingredients, such as glycyrrhetinic acid [84] and vitamin E [93]. The CN-NL complexed capsules can be formulated in the form of suspension and release the active ingredient on both scalp and hair, by a simple soft massage, or as dry nanostructured tissues that can be applied on the hair for a longer time and result in a longer time-release of the active ingredients [92]. Besides its role as a carrier, lignin can also act as an active ingredient due to its antibacterial, antioxidant, anti-inflammatory, regenerative, and sun-protective activities [94]. The −OH groups of nanolignin can also contribute to reestablish broken disulfide and hydrogen bonds by helping in crosslinking neighboring keratin molecules [11].

Although studies on lignin as an active agent in hair products are scarce, its application in cosmetic products for skin care has been proposed, especially in sunscreens. Lignin contains several functional groups that absorb UV radiation, making it a great candidate to be used as a natural UV-protector agent in sunscreens or creams [95]. Adding small amounts of lignin into commercial sunscreen products leads to a significant enhancement in their ultraviolet absorbance [96]. The sunscreen performance of lignin can be further improved when using lignin colloidal spheres. Spherical lignin has better UV-blocking performance than that of original lignin, and it increases inversely with the size of the colloidal spheres [83]. Besides its sun-protective function, lignin has a high phenolic content and antioxidant activity, presenting tyrosinase-inhibitory properties [97]. Thus, lignin shows a great potential as a bioactive multi-functional cosmetic ingredient for skin care applications offering sun-protection, anti-ageing and skin-whitening properties [97].

These unique features make lignin a very appealing raw material for the synthesis of natural ingredients for hair care formulations. The study of the conditioning properties of lignin and lignin derivatives would contribute to the advancement and expansion the availability of natural-based ingredients for hair care formulations.

## 5. Concluding Remarks

Hair is constantly exposed to thermal, chemical, or environmental degradation that can leave the hair with an unpleasant feeling and appearance. To improve the physical and aesthetic properties of the hair, conditioning agents are incorporated into hair care formulations, such as shampoos and hair conditioners. These agents are responsible for reducing the friction between fibers, smoothing the hair surface, and restoring hair gloss and softness. Different types of hair conditioners, i.e., cationic, film-forming, and protein-based conditioners, contain compounds with different chemical structures and present different mechanisms of action. The choice of the most suitable conditioner depends on the desired effect and damage degree. In addition to the product’s efficiency, its origin and environmental impact are becoming concerns for many consumers that are driving their preferences towards more sustainable and natural-based formulations. Protein-based conditioners, for example, can be easily prepared from natural ingredients such as soy, rice, quinoa, or other plant-derived proteins. On the other hand, cationic conditioners, designed to treat extremely damaged hair by electrostatically interacting with negatively charged hair fibers, reducing frizz, and making the hair smooth, are frequently produced from non-renewable resources and pose some risks to aquatic life. Although some efforts are being made to develop new cationic conditioning agents, such as esterquats or quaternized wheat protein hydrolysates, more alternatives are still needed. Therefore, there is an increased need to find new components that can fulfill consumer expectations both in terms of efficiency and environmental impact, especially in the case of cationic conditioners. Concepts such as “sustainability”, “circular economy”, and “zero waste” are also paving the way towards a new era that treats residues as useful resources to produce new materials of added value. Biopolymers, such as lignin, are excellent candidates to be used in bio-based formulations. Lignin is a natural polyphenol that has been reported as a multi-functional cosmetic ingredient for hair and skin care, offering antibacterial, antioxidant, regenerative, and sun-protective activities. Despite its high potential for the development of novel value-added biomaterials, lignin is highly underutilized, being mostly burned as low-grade fuel.

We believe that biomolecules such as cellulose and lignin have great potential as future ingredients for the design of new, efficient, and sustainable hair conditioning formulations. These new conditioning agents should be able to efficiently repair damaged hair and not compromise the environment or consumer health.

## Figures and Tables

**Figure 1 polymers-15-00608-f001:**
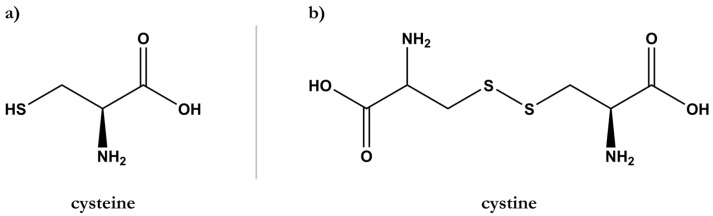
Chemical structures of (**a**) cysteine amino acid and (**b**) cystine dimer formed by disulfide bonds between two cysteine molecules.

**Figure 2 polymers-15-00608-f002:**
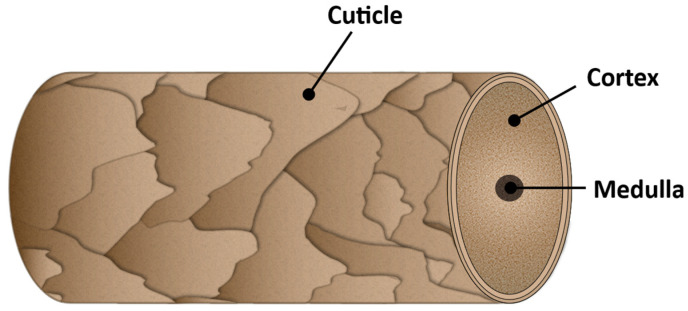
Schematic representation of the hair shaft structure.

**Figure 3 polymers-15-00608-f003:**
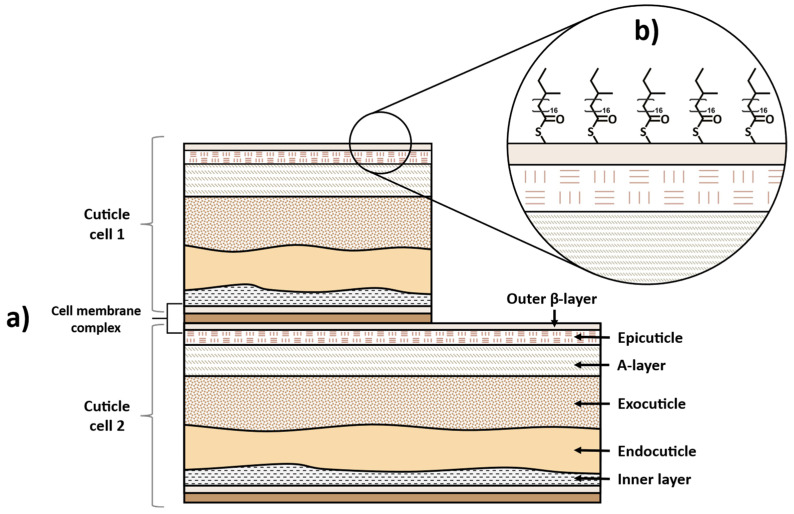
Schematic representation of (**a**) structure of hair cuticle cell and (**b**) lipid layer of 18-methyleicosanoic acid (18-MEA) covalently linked to the epicuticle outer surface via thioester linkages. Adapted from reference [13].

**Figure 4 polymers-15-00608-f004:**
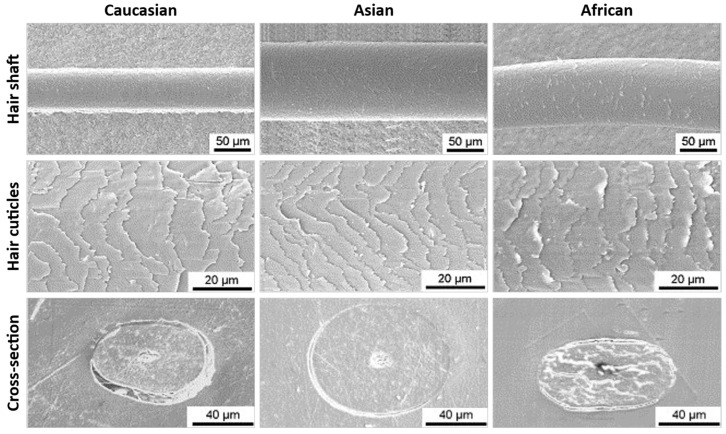
Comparison of hair fibers surface and cross-section from Caucasian, Asian and African people by SEM. Adapted from reference [23] with permission from Elsevier.

**Figure 5 polymers-15-00608-f005:**
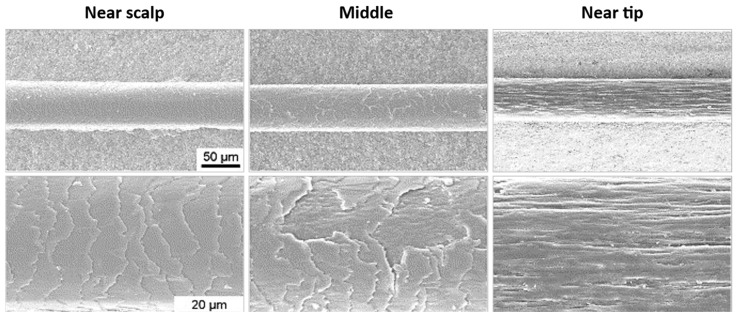
SEM images of a virgin hair fiber at three locations: near the scalp, in the middle of the fiber and near the tip. Adapted from reference [23] with permission from Elsevier.

**Figure 6 polymers-15-00608-f006:**
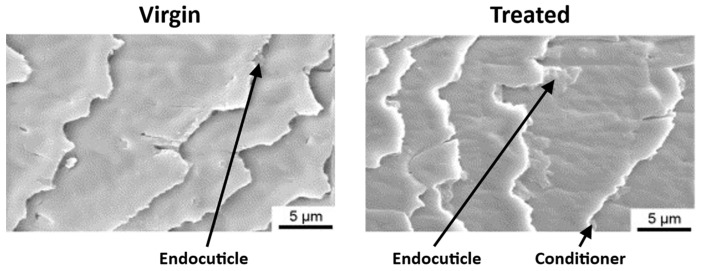
SEM images of virgin and treated hair. Adapted from reference [23] with permission from Elsevier.

**Figure 7 polymers-15-00608-f007:**
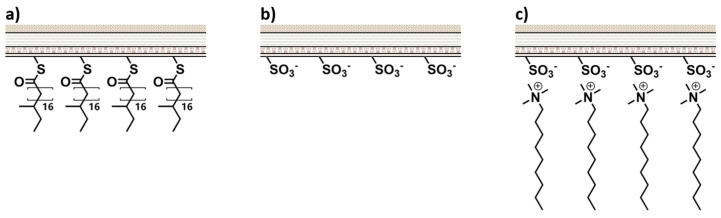
Repairing action of cationic conditioners: (**a**) undamaged hair covered with its natural hydrophobic protective F-layer; (**b**) negatively charge damaged hair without the F-layer; (**c**) hair repaired with conditioner, the cationic surfactant adsorbs to the surface. Adapted from reference [44] with permission from Elsevier.

**Figure 8 polymers-15-00608-f008:**
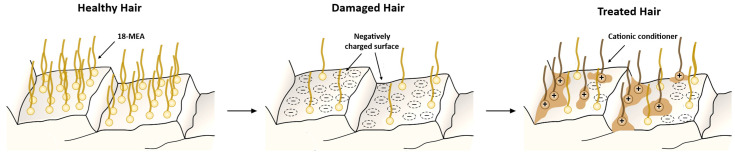
Negatively charged hair and the deposition of positively charged conditioner on the cuticle surface. Adapted from reference [50].

**Figure 9 polymers-15-00608-f009:**
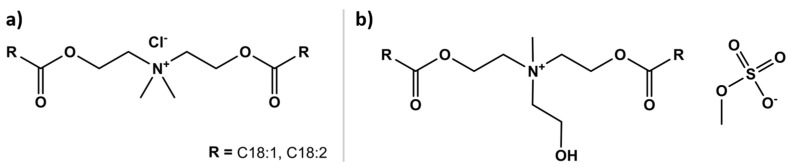
Chemical structure of two commercially available plant-based esterquats derived from sunflower seed oil used in hair conditioners formulations: (**a**) disunfloweroylethyl dimonium chloride and (**b**) dioleoylethyl hydroxyethylmonium methosulfate.

**Figure 10 polymers-15-00608-f010:**
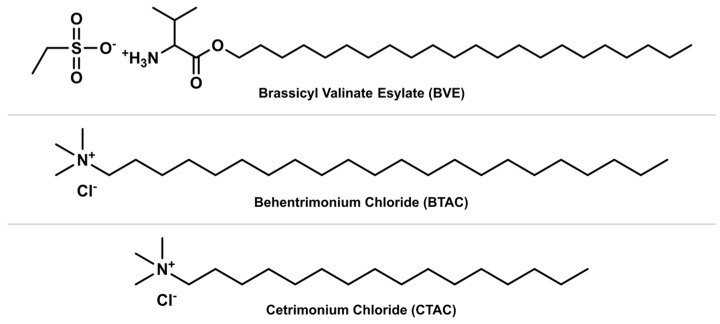
Chemical structures of cationic amino lipid surfactant brassicyl valinate esylate (BVE) and alkyl quaternary ammonium surfactants behentrimonium chloride (BTAC) and cetrimonium chloride (CTAC).

**Table 1 polymers-15-00608-t001:** Types of hair conditioner products and their intended application.

Classification	Application	Conditioning Agents Used
Shampoos “2 in 1”	Shampoo with dual function—cleaning and conditioning. Conditioning additives are incorporated to impart manageability, gloss, and antistatic properties to the hair. Recommended for dry, damaged, or chemically treated hair.	Hydrolyzed protein, silicones, glycerin, polyvinylpyrrolidone or quaternary conditioners.
Instant conditioners	Intended to be applied after shampooing and rinsed after a short period of time (around 5 min). Ideal for daily use in slightly damaged hair to reduce the effect of shampoo and improve daily manageability.	Quaternary conditioners, such as behentrimonium chloride and stearalkonium chloride.
Deep/intensive conditioners	More concentrated than instant conditioners and should be left on the hair for 20 to 30 min. Recommended for extremely dry hair or before chemical treatment, such as coloring and waving.	Higher amounts of quaternary conditioners in addition to proteins.
Leave-in conditioners	Designed to be applied after shampooing and conditioning, and not rinsed out. These products can be applied daily in wet or dry hair and are ideal for preventing damage from routine grooming.	Silicones, oils, polyvinylpyrrolidone or other film-forming agents.
Blow drying conditioners	They are applied to towel-dried hair before blow-drying and styling and may provide photoprotection and prevent heat damage. Useful for people with fine hair and excessive scalp sebum.	Same agents as instant conditioners but do not contain oil.
Hair thickeners	They coat the hair shaft, increasing their diameter and giving the illusion of thick hair. They usually contain proteins as conditioners and are also applied to towel-dried hair before styling.	Film-forming agents, such as silicones or hydrolyzed proteins, such as keratin K31.

**Table 2 polymers-15-00608-t002:** Classification of hair conditioners based on their composition.

Classification	Mode of Action	Ingredients
Cationic conditioners	Act by neutralization of the negative charges of damaged hair by deposition of positively charged molecules on the hair surface. The softness and smooth appearance of the hair are achieved by the reduction of static electricity of the cuticles. Excellent for chemically processed hair.	Quaternary ammonium compounds: cetrimonium chloride, stearalkonium chloride, etc.
Film-forming conditioners	Act by deposition of polymers that form a film that fills the defects in cuticles surface and coat the hair shaft, restoring its softness and shine. They can also be positively charged and reduce the static electricity of anionic damaged hair shaft.	Film-forming agents, such as polyvinylpyrrolidone (PVP), silicones and oils.
Protein-based conditioners	Contain amino acids and small polypeptide fragments of hydrolyzed proteins that can penetrate the hair shaft and repair the damaged hair by restoring the lost proteins and improve the hair’s strength. The excess proteins are rinsed out when washing the hair, so their effect is only temporary.	Many different protein sources: animal protein, eggs, placenta, collagen, keratin, beer, among others.

**Table 3 polymers-15-00608-t003:** Main natural polymers and their derivatives used in hair conditioning products. Data collected from reference [76].

Name	Function	Examples of Derivatives Used in Cosmetics [INCI]
Starch (maize, potato, tapioca…)	Conditioner; softening agent; thickener	Aluminium Starch Octenyl Succinate Distarch Phosphate Hydroxypropyl Starch Phosphate Starch Hydroxypropyltrimonium Chloride.
Galactomannan gums (guar and locust bean)	Film former; stabilizer; thickener	Hydroxypropyl Guar Guar Hydroxypropyltrimonium Chloride Locust Bean Hydroxypropyltrimonium Chloride
Hydrolyzed wheat	Conditioner; film former; antistatic; tensor	Hydrolyzed Wheat Protein/Dimethicone PEG-7 Acetate Hydrolyzed Wheat Protein/PEG-20 Acetate Copolymer Hydrolyzed Wheat Protein PG-Propyl Silanetriol
Hydrolyzed keratin	Conditioner; moisturizer	Cocodimonium Hydroxypropyl Hydrolyzed Keratin Hydrolyzed Keratin PG-Propyl Methylsilanediol
Collagen, gelatin, and hydrolyzed collagen	Conditioner; film former; moisturizer; hydrating;	Cocodimonium Hydroxypropyl Hydrolyzed Collagen Laurdimonium Hydroxypropyl Hydrolyzed Collagen Isostearoyl Hydrolyzed Collagen
Chitosan, hydrolyzed chitosan and chitin	Conditioner; film former; thickener; chelating agent; hydrating	Chitosan Lactate Chitosan PCA
Hydrolyzed silk	Hair conditioner; antistatic; humectants	Hydrolyzed Silk PG-Propyl Methylsilanediol Sodium Lauroyl Hydrolyzed Silk
Cellulose derivatives	Film former; emulsion stabilizer; viscosity control	Cetyl Hydroxyethylcellulose Hydroxyethylcellulose Hydroxypropylcellulose
Cationic cellulose derivatives	Antistatic, film forming	Polyquaternium-10 (Cationic hydroxyethyl cellulose)

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
