# Peer review of "On Hair Care Physicochemistry: From Structure and Degradation to Novel Biobased Conditioning Agents"

_polymers, 2023, doi:10.3390/polym15030608_

Round 1

Reviewer 1 Report

Comments:

1.         page 1, line 39 the authors write: "These products are responsible for the removal of dust..." I suggest supplementing with: "These products are responsible for the removal of dust, residues from styling cosmetics, ..."

2.  page 8, lines 316-320 the authors write: "Although some degree of conditioning can be achieved with the incorporation of conditioning additives in shampoo formulations, their effect is not as pronounced as in a hair conditioner formulation, possibly due to incompatibilities between the components, in particular, anionic surfactant cleaners and cationic conditioners [36]." I agree with this statement. However, I would suggest that the sentence be supplemented with the phrase : "which may react with each other and, as a result, create complexes insoluble in water".

3. page 16, lines: 621, 622, 637 I suggest you do not use the trade names of the compounds (STEPANQUAT® Helia, STEPANQUAT®Soleil, AminoSensyl™). This is an advertisement for these commercial products. Replace them in the text with INCI. Similarly, use INCI in the headings of Figure 9.

4.         page 17, Table 3,  first-right column The title of the column "Examples of derivatives used in cosmetics" should be completed  „Examples of derivatives used in cosmetics [INCI]”. Listing all cosmetic ingredients International Nomenclature of Ingdidients (INCI) should be used.  Throughout the table, name the substances correctly: each term of the ingredient name should be given using capital letters, e.g.” Aluminium starch octenyl succinate” it should be written:  „Aluminium Starch Octenyl  Succinate”, etc.

5. page 17, Table 3,  first-right column , abbreviations may not be used in INCI names.  e.g. „ Hydr. wheat protein peg-20 acetate copolymer”.   Llease use the full phrase:  Hydrogenated. All name of the compound/mixture should be corrected  „Hydrogenated  Wheat Protein PEG-20 Acetate Copolymer”.

6. page 17-18, apply the comments from points  4 and 5 to Table  3.

Author Response

We deeply appreciate and thank the reviewer’s careful analysis of our work. We have carefully considered the comments and tried our best to address every one of them.

  1. page 1, line 39 the authors write: "These products are responsible for the removal of dust..." I suggest supplementing with: "These products are responsible for the removal of dust, residues from styling cosmetics, ..."

The sentence was revised accordingly.

  1. page 8, lines 316-320 the authors write: "Although some degree of conditioning can be achieved with the incorporation of conditioning additives in shampoo formulations, their effect is not as pronounced as in a hair conditioner formulation, possibly due to incompatibilities between the components, in particular, anionic surfactant cleaners and cationic conditioners [36]." I agree with this statement. However, I would suggest that the sentence be supplemented with the phrase: "which may react with each other and, as a result, create complexes insoluble in water".

The sentence was revised accordingly.

  1. page 16, lines: 621, 622, 637 I suggest you do not use the trade names of the compounds (STEPANQUAT® Helia, STEPANQUAT®Soleil, AminoSensyl™). This is an advertisement for these commercial products. Replace them in the text with INCI. Similarly, use INCI in the headings of Figure 9.

Thank you very much for pointing this out. We have revised the main text and figures content, and all mentions to commercial names were replaced/deleted.

  1. page 17, Table 3, first-right column The title of the column "Examples of derivatives used in cosmetics" should be completed  „Examples of derivatives used in cosmetics [INCI]”. Listing all cosmetic ingredients International Nomenclature of Ingdidients (INCI) should be used.  Throughout the table, name the substances correctly: each term of the ingredient name should be given using capital letters, e.g.” Aluminium starch octenyl succinate” it should be written:  „Aluminium Starch Octenyl  Succinate”, etc.

Thank you for your comment. The suggestions were followed and the column heading and compounds names were revised accordingly.

  1. page 17, Table 3,  first-right column , abbreviations may not be used in INCI names.  e.g. „ Hydr. wheat protein peg-20 acetate copolymer”.   Llease use the full phrase:  Hydrogenated. All name of the compound/mixture should be corrected  „Hydrogenated  Wheat Protein PEG-20 Acetate Copolymer”.

Thank you for your suggestion. All names listed in Table 3 were carefully reviewed and all abbreviations were replaced by the INCI names.

  1. page 17-18, apply the comments from points 4 and 5 to Table 3.

The entire compounds list was carefully reviewed and revised according to the previous comments.

Reviewer 2 Report

Over the past year, I have reviewed more than one dozen articles for publication.  This article stands out.  The article is historically comprehensive and the subject material addressing "the circular economy and zero-waste approaches" to hair conditioning is vital to the field of modern hair care.  I found that reading it made me excited, almost like a suspenseful novel, reading each new paragraph with great anticipation.  The authors never disappointed me as the article is not only well written, but comprehensive and well referenced.  I believe, that this article will become a 'roadmap' for producing "future ingredients for the design of new efficient and sustainable hair conditioning formulations"  I commend the authors for an excellent well-written article. 

Author Response

Over the past year, I have reviewed more than one dozen articles for publication.  This article stands out.  The article is historically comprehensive and the subject material addressing "the circular economy and zero-waste approaches" to hair conditioning is vital to the field of modern hair care.  I found that reading it made me excited, almost like a suspenseful novel, reading each new paragraph with great anticipation.  The authors never disappointed me as the article is not only well written, but comprehensive and well referenced.  I believe, that this article will become a 'roadmap' for producing "future ingredients for the design of new efficient and sustainable hair conditioning formulations"  I commend the authors for an excellent well-written article. 

We deeply appreciate and thank the reviewer’s careful analysis of our work and the kind comments and positive feedback. We are grateful that the reviewer has positively recognized our effort in the elaboration of this work.